# Biosynthesis of DNA-Alkylating Antitumor Natural Products

**DOI:** 10.3390/molecules27196387

**Published:** 2022-09-27

**Authors:** Qiu-Yue Nie, Yu Hu, Xian-Feng Hou, Gong-Li Tang

**Affiliations:** 1State Key Laboratory of Bioorganic and Natural Products Chemistry, Center for Excellence in Molecular Synthesis, Shanghai Institute of Organic Chemistry, University of Chinese Academy of Sciences, Chinese Academy of Sciences, 345 Lingling Road, Shanghai 200032, China; 2School of Chemistry and Materials Science, Hangzhou Institute for Advanced Study, University of Chinese Academy of Sciences, 1 Sub-lane Xiangshan, Hangzhou 310024, China

**Keywords:** biosynthesis, DNA alkylating agents, antitumor natural products, resistance

## Abstract

DNA-alkylating natural products play an important role in drug development due to their significant antitumor activities. They usually show high affinity with DNA through different mechanisms with the aid of their unique scaffold and highly active functional groups. Therefore, the biosynthesis of these natural products has been extensively studied, especially the construction of their pharmacophores. Meanwhile, their producing strains have evolved corresponding self-resistance strategies to protect themselves. To further promote the functional characterization of their biosynthetic pathways and lay the foundation for the discovery and rational design of DNA alkylating agents, we summarize herein the progress of research into DNA-alkylating antitumor natural products, including their biosynthesis, modes of action, and auto-resistance mechanisms.

## 1. Introduction

Natural products (NPs) are an important source of pharmaceuticals due to their diverse bioactivities [1]. Since DNA is essential for living organisms, DNA-targeting NPs, which usually function as carcinogenesis or cancer treatment, constitute an indispensable family of bioactive NPs [2,3]. Although the genotoxic metabolite colibactin, produced by human gut bacteria, is shown to cause colorectal cancer by alkylating DNA to generate DNA mutation [4,5,6], some DNA-targeting NPs are applied in chemotherapy. They can interact with specific DNA duplex structures and cause DNA damage via different modes of action [7]. One of the mechanisms is the cleavage of DNA through inducing the production of radical DNA by redox reactions or nucleophilic addition. Broad anti-cancer antibiotic bleomycin (BLM) can be transformed to HOO-Fe(III)-BLM in the presence of Fe/O_2_ to damage DNA [8]. The enediyne-containing NPs dynemicin A and calicheamicin can generate biradical intermediates to cleave DNA activated by reducing the quinone moiety and the attack of a thiol, respectively [9,10,11,12]. Some chemicals, such as streptozotocin, conduct the methylation of DNA [13,14]. Additionally, another family of DNA-targeting antitumor agents can alkylate DNA in situ with covalent bonds. They can directly react with DNA using highly active functional groups such as epoxide, cyclopropane, and aziridine to form bulky DNA adducts [15,16]. Furthermore, as a result of the potent cytotoxicity of DNA-alkylating NPs, it is preferable for their producers to possess resistant genes located in biosynthetic gene clusters (BGCs) to protect themselves. BGCs-associated self-resistance is mainly achieved through excision of the abnormal base, degradation of active functional groups, and the binding or transport of toxins [17]. 

The biosynthesis and resistance of radical-based DNA damage agents, including BLMs and enediynes, have already been well reviewed [18]; herein, we mainly discuss DNA-alkylating (except DNA-methylation) antitumor NPs, including their modes of action, BGC-associated self-resistance, and biosynthetic pathways, especially the construction of their highly active groups as a warhead. 

## 2. Spirocyclopropane-Containing Cyclohexadienone Natural Products

The spirocyclopropylcyclohexadienone family, including yatakemycin (YTM, **1**), CC-1065 (**2**), and duocarmycin SA (**3**), all contain a highly active cyclopropane moiety and exhibit potent antitumor activities (Figure 1) [19,20,21,22]. Duocarmycin-based antibody-drug conjugates (ADC, SYD985 (**4**), and MDX-1203 (**5**)) have entered clinical trials for the treatment of specific cancers as prodrugs (Figure 1) [23,24]. They can selectively bind AT-rich regions in the DNA minor groove by non-covalent interaction, then form a covalent bond with DNA in which the cyclopropanol group is attacked by the N-3 of adenine (Figure 2A) [25]. The YTM-producer was first identified as protecting itself with DNA glycosylases YtkR2 through the base-excision repair mechanism (Figure 2A). The homologous enzyme C10R5 exhibited a similar function in the CC-1065-producing strain. [26] Because the cyclopropane warhead exhibits strong potency, additional self-protection of their hosts can also be achieved by the cleavage of this moiety. A GyrI-like protein was verified to hydrolyze the cyclopropane moiety in YTM and CC-1065 to facilitate detoxification (Figure 2B) [27,28,29]. 

The benzodipyrrole scaffold in CC-1065 was derived from serine, methionine, and tyrosine-derived DOPA, and was revealed by isotopic feeding experiments (Figure 3) [30,31]. Wu et al. proposed the possible biosynthetic pathway of CC-1065. Tyrosine was first oxidized to DOPA which underwent intramolecular cyclization to afford **10**. Next, **11** produced by the combination of serine and **10** was decarboxylated and cyclized to yield **13** which was further modified to form three different types of building blocks (**17**, **18**, and **19**). The assembly of these building blocks generated the final core structure **21** (Figure 4).

Strategies for the incorporation of cyclopropane have long fascinated chemists, since it is an important synthetic building block and a common pharmacophoric group. The chemical synthesis of cyclopropane in this family of NPs was mainly achieved by nucleophilic cyclopropanation [32]. In the biosynthesis of CC-1065, Jin et al. reported that a two-component cyclopropanase system consisting of a HemN-like radical S-adenosylmethionine (SAM) enzyme C10P and a methyltransferase C10Q was responsible for generating the essential cyclopropane moiety involving a unique enzymatic mechanism (Figure 5) [33,34]. To explain in detail, the highly active SAM methylene radical attacks the C-11 position of **22** to generate the radical intermediate **23**, which subsequently abstracts hydrogen to yield the SAM-substrate adduct **24**. Following this, the deprotonation of the phenolic hydroxyl group in virtue of His-138 residue in C10Q induced S_N_2 cyclopropanation to produce CC-1065 with S-adenosylhomocysteine (SAH) as the leaving group. Additionally, **24** could also be converted to **25** by non-enzymatic reaction with the release of SAH, and the following isomerization produces the methylated compound **26**.

## 3. DNA-Alkylating Natural Products with Heterocyclic Propane as Pharmacophore

### 3.1. Pluramycins

As an important family of NPs, type II polyketides display various structurally diverse biological activities [35,36]. Anthracycline compounds such as daunomycin and nogalamycin exhibit antitumor activities by intercalating into grooves of DNA, while most of these compounds are unable to form a covalent bond with duplex DNA [37,38]. Nevertheless, pluramycin antibiotics including hedamycin (**27**) and altromycin B (**28**) (Figure 6A), which usually contain an epoxide moiety, can intercalate and alkylate DNA simultaneously. Similar to daunomycin, their anthraquinone ring was characterized as intercalating into DNA and binding saccharides in the minor or major groove, thereby contributing to the stabilization of the drug–DNA complex [39,40,41]. Furthermore, their epoxides could be opened via nucleophilic attack of N-7 of guanine, resulting in the formation of an adduct by covalent bond (Figure 6B).

Biosynthetically, the epoxides in hedamycin were formed on its non-acetyl starter unit generated by two separate type I polyketide synthases (PKSs, HedT, and HedU). HedU was proposed to catalyze two rounds of chain elongation employing the acetyl starter unit provided by HedT (Figure 7) [42,43,44]. The obtained unsaturated 2,4-hexadienyl unit was then transferred to the downstream type II PKS to produce the aromatic precursor. The following oxidation of the C2-alkyl side in intermediate **30** afforded the epoxide intermediate **31** which was further modified by methyltransferase and two C-glycosyltransferases to yield hedamycin (Figure 7). 

Trioxacarcins (TXNs), firstly isolated from *Streptomyces bottropensis* NRRL 12051 in 1981, exhibit extraordinary antibacterial, antimalarial, and antitumor activities [45]. Among the TXNs, TXN A (**35**) showed the most potent antitumor activities with sub-nanomolar IC_50_ values against various cancer cell lines (Figure 8A). The unique fused spiro-epoxide of TXN A is essential for its bioactivities because the epoxide can react with N-7 of guanine to form a DNA–TXN A complex (Figure 8B). The crystal structure of this complex revealed that glycosyl groups at C-4 and C-13 were docked with the minor and major groove, respectively [46]. It also displayed an unexpected flipping out of the base at the intercalation site, which might be important for DNA–protein interaction. Recently, the study of the TXN analog LL-D49194α1 (**36**) showed that the deglycosylated compounds (**37** and **38**) exhibited more potent anticancer activities than **36,** possibly suggesting a new mode of interaction with DNA (Figure 8A) [47]. Furthermore, the DNA–TXN complex (**39**) could be cleaved to yield gutingimycin (**40**) involving a self-resistance mechanism of an excising base (Figure 8B) [48,49]. Recently, four DNA glycosylases, TxnU2, TxnU4, LldU1, and LldU5, were reported to be responsible for excising the intercalated guanine adducts [50].

Based on the work of Zhang et al., TXNs were also biosynthesized from anthraquinone-intermediate parimycin (**47**), similarly to the precursor of hedamycin, as revealed by gene-deletion experiments [45]. According to isotope-labelled precursor feeding experiments, the unusual starter unit 2-methylbutyryl of TXNs was derived from L-isoleucine through transamination. After a series of modifications, including condensation with acetyl-CoA and decarboxylation, this starter unit was incorporated into the polyketide chain in virtue of KSIII (Figure 9). The formation and subsequent cyclization of the polyketide chain provided intermediate **46,** whose pyrone ring was formed by a CalC-like protein, TxnO9 [51]. The decarboxylated intermediate **47** underwent complex tailoring steps to afford intermediate **51** with a unique spiro-epoxide structure, but the specific enzymatic process and mechanism remained uncharacterized. Following that, the methylation of **51** at C-4 and C-13 yielded **52,** whose C4-sugar was finally acetylated by the membrane-bound *O*-acetyltransferase TxnB11 to form **35** (Figure 9) [52]. Unlike TXNs, the C-16 and C-4 of **45** were glycosylated and methylated to produce LL-D49194α1. 

### 3.2. Mitomycins

Mitomycins (MMs, such as MMA, B, and C) are antitumor NPs discovered in *Streptomyces.* They all contain the quinone backbone and a unique azabicycle moiety (Figure 10A) [18,53]. Among these compounds, MMC has been used as a chemotherapeutic agent in the clinic for more than five decades. MMC can form inter-strand and intra-strand cross-linking with DNA at the selective sequence (5’-CG-3’) and resides in the minor groove [54]. Other compounds of the mitomycin family, such as FR900482 and FR66979, also showed potent DNA cross-linking activity as well as bioactivities against cancer cell lines (Figure 10A). FR900482 was superior to MMC in both efficacy and safety [55]. 

The mode of action of MMs is well studied. Firstly, a reductive pathway is required to activate the quinone moiety of MMC by either enzymatic or chemical means to form the hydroquinone intermediate **60** [18,53]. Subsequent elimination of methanol in **60** affords **61** which undergoes tautomerization and the ring-open reaction of aziridine ring to yield **65** (Figure 10B). The N-2 of guanine attacks the C-1 position to generate the DNA–compound complex, then the departure of carbamate produces the iminium intermediate **69,** which is attached by the second guanine of DNA in the same way to form **71**. Furthermore, the first reductive activation could be inhibited by a FAD-dependent oxidoreductase MCRA (encoded by *mcrA*) which enables the oxidization of the hydroquinone form to the quinone form to confer self-resistance [56,57,58,59]. Although the alkylating mechanism of FR900482 is similar to that of MMC, it is activated by cleaving the N-O bond to form **59** (Figure 10B). 

Since these compounds possess excellent bioactivities and the common pharmacophoric group azabicycle, their synthesis has attracted extensive attention. In chemical synthesis, the azabicycle moiety of MMs is installed from benzazocane intermediates via intramolecular substitution [60], but their biosynthetic pathways are still not well elucidated. According to isotopic precursors feeding experiments conducted by Hornemann et al., the origins of the *O*-methyl group and the carbamate were methionine and L-citrulline, respectively, while the mitosane core was derived from 3-amino-5-hydroxybenzoic acid (AHBA, **81**) and glucosamine [61,62]. The precursor AHBA was formed via the amino-shikimate pathway related to rifamycin and kanosamine biosynthesis [63,64]. After the formation of AHBA, it was firstly activated by acyl AMP-ligase MitE and was then loaded onto acyl carrier protein (ACP) MmcB (Figure 11). The glycosyltransferase MitB was verified to catalyze the glycosylation of AHBA-MmcB with UDP-GlcNAc [65,66,67]. Recently, Wang et al. traced all the ACP-channelled MM intermediates indicating that AHBA-MmcB-GlcNAc intermediate **85** should undergo the deacetylation by MitC to form **86** which was further transformed to **88** by MitF and MitD [68]. The epoxide intermediate might be cyclized to provide benzazocine **92**. **92** then underwent oxidation and several uncovered modifications to generate hydroxyquinone intermediate **95** which was methylated to afford MMA, the direct precursor of MMC [69]. Sherman and co-workers also identified a methyltransferase MitM which methylated the nitrogen of aziridine in MMA rather than MMC to yield MMF (**96**) [70]. In addition, the epoxide of **88** could be opened by the nucleophilic attack to afford **89** which was the precursor of the MMs with α-C9. Moreover, the oxidation of the aniline amine in **89** facilitated forming the core structure of FR900482. 

### 3.3. Azinomycins

The antitumor antibiotics azinomycin A (**98**) and B (**99**) contain naphthoic acid (NPA) moiety, epoxide, and azabicyclohexane ring which all contribute to alkylating DNA (Figure 12) [71]. The electrophilic epoxide and aziridine can both be attacked by N-7 of guanine and the latter can even be opened by N-7 of adenine, leading to the formation of interstrand DNA cross-links (Figure 12) [72,73]. NPA moiety also plays an important role in the DNA alkylating activity by virtue of non-covalent interactions [74]. In 2011, the aminoglycoside transferase AziR was identified to mediate the self-resistance of azinomycin and reduce the DNA damage via binding azinomycin. Recently, a novel DNA glycosylase Orf1 and an endonuclease AziN were reported to repair the DNA damage to achieve self-protection [75,76,77,78].

Previous isotope-labelled precursor feeding experiments revealed that the epoxy moiety, the azabicyclic fragment, and the terminal part in azinomycin B were derived from acetyl-CoA, valine, glutamic, and threonine, respectively (Figure 13) [79,80]. The feeding experiments with isotopically labelled substrates showed that 3-methyl-2-oxobutenoate (**105**) was incorporated into the azinomycin epoxide as the penultimate precursor (Figure 14A). The formation of **105** was achieved by oxidation, transamination as well as dehydration beginning with L-valine. Even so, the exact timing of forming epoxy amide remains unclear up to now [81].

In the biosynthesis of this class of non-ribosomal peptide-polyketide hybrid compounds, iterative type I PKS AziB catalysed the formation of 5-methyl-NPA (**100**) which was further transformed to 3-methoxy-5-methyl-NPA (**102**) by a P450 hydroxylase AziB1 and the *O*-methyltransferase AziB2 (Figure 14B) [82]. The first building block **102** was activated by the distinct adenylation (A) domain of the di-domain non-ribosomal peptide synthetase (NRPS) AziA1 to initiate the backbone formation of azinomycins [83,84].

The azabicycle moiety was constructed from 3,4-epoxypiperidine derivatives via spontaneously intramolecular substitution in chemical synthesis [85]. Watanabe and co-workers unraveled the biosynthetic pathway of the azabicyclic fragment in azinomycin, wherein the glutamic acid was initially acetylated at the amino group by *N*-acetyltransferase AziC2 to form *N*-acetyl glutamate **106**. The *N*-acetyl glutamate kinase AziC3 subsequently phosphorylated the carboxyl to afford the *N*-acetyl-glutamyl 5-phosphate (**107**) which was subsequently reduced to *N*-acetyl-glutamate-5-semialdehyde (**108**) by an *N*-acetyl-γ-glutamate phosphate reductase AziC4 (Figure 14C), and the key two-carbon extension on aldehyde intermediate catalyzed by the transketolase AziC5/C6 afforded **110** which was further converted to the acetylated nonproteinogenic amino acid diamino-dihydroxy-heptanoic acid (DADH, **111**) by an aminotransferase (AziC1 or AziC7) [86,87,88]. Recently, Kurosawa et al. demonstrated that **111** could be further sulphated to **112** and the sulfate group in **112** was finally attacked by the ortho amino group to form the aziridine ring intermediate **113** which may be subsequently acetylated and cyclized to form the azabicyclic fragment **114** (Figure 14C) [89]. Additionally, glutamic acid might be firstly activated by the amino-group carrier protein (AmCP) and was further modified to produce DADH which was then introduced into the azabicyclic structure according to a recent study about the biosynthesis of vazabitide A [90]. Moreover, the enol in the final building block of azinomycin B was generated by the oxidation of L-threonine, while the decarboxylation of the intermediate **116** afforded the aminoacetone **117** in azinomycin A [91].

## 4. DNA-Alkylating Natural Products with Imine as Warheads

### 4.1. Pyrrolobenzodiazepines

Antitumor antibiotics pyrrolobenzodiazepines (PDBs), including anthramycin, sibiromycin, and tomaymycin, all contain three parts: anthranilate, diazepine, and hydropyrrole (Figure 15A) [92,93]. The imine in the diazepine can be attacked by N-2 of guanine to form a stable covalent bond resulting in inhibiting DNA synthesis (Figure 15C) [94]. Moreover, the crystal structure of the anthramycin-DNA complex indicates that the *S*-configuration of C-11a make it suitable for docking in the minor groove of DNA [95,96]. In addition, a PDB dimer, SJG-136 (**121**), which has completed the phase II clinical trial for treating leukemia and ovarian cancer, can form DNA inter-strand and intra-strand cross-linking of DNA (Figure 15B) [97,98].

Based on previous feeding experiments, L-methionine, L-tyrosine, and L-tryptophan were supposed to be the biosynthetic precursors of pyrrolobenzodiazepines (Figure 15A) [99]. Like lincomycin biosynthesis, L-dopa from L-tyrosine is cleaved to yield semialdehyde intermediate **123**, followed by intramolecular cyclization to form dihydropyrrole intermediate **124** (Figure 16A) [100]. The decarboxylation of intermediate **124** generates **125,** which is further converted into a variety of dihydropyrrole precursors (**130**-**132**) to be introduced into PDB biosynthesis [101,102,103]. For the biosynthesis of hydroxyanthranilic acid intermediates, L-tryptophan was firstly degraded to L-kynurenine (**134**), the biosynthetic precursor of important NPs including actinomycin, quinolobactin, and daptomycin [104,105,106]. Following this, **134** goes through three continuous tailoring steps mediated by monooxygenase, kynurenine hydrolase, and methyltransferase, respectively, to generate 3-hydroxyl-4-methyl-anthranilic acid (**137**), which is subsequently introduced into the anthramycin or oxidized to be the precursor (**138**) of sibiromycin (Figure 16B) [107]. Two NRPSs containing two (A-PCP) and four domains (C-A-PCP-RE), respectively, are responsible for the formation of the amide bond between two building blocks, and the release and intermolecular cyclization of the chain to afford **139**, which is dehydrated to form the final compounds with imine moieties (Figure 16C) [108]. 

### 4.2. Tetrahydroisoquinolines

Tetrahydroisoquinoline NPs are mainly classified into three subfamilies composed of the saframycin (SFM) family including SFM A (**141**) and ET-743 (**143**) and the naphthyridinomycin (NDM) family, including NDM A (**140**), as well as the quinocarcin family compounds (Figure 17A) [109]. Most SFMs exhibit inhibitory activity against cancer cell lines by alkylating DNA. ET-743, produced by the bacterial symbiont *Candidatus Endoecteinascidia frumentensis*, displayed the most potent antitumor activities and has been used clinically to treat ovarian neoplasms and sarcomas [110]. Two mechanisms of SFM A for alkylating DNA were reported. One way was the formation of iminium intermediate **148** through reduction of the quinone moiety (Figure 17B) [111]. In the other way, the ortho-position nitrogen could directly promote the departure of the functional group in C-21 to yield the iminium intermediate **144**. Both **144** and **148** could be attacked by nucleophilic N-2 residue of guanine in GC-rich regions of DNA to form the DNA–drug complex (Figure 17B) [112]. Furthermore, a FAD-binding oxidoreductase NapU encoded in BGC of NDM was reported to activate and inactivate the matured prodrug by extracellular oxidation conferring self-protection [113]. Recently, a short-chain dehydrogenase NapW mediated the reduction of the hemiaminal pharmacophore, implicating another level of the self-resistance mechanism of the tetrahydroisoquinoline family [114].

Isotope-labeled precursor feeding experiments have revealed that the skeleton structure of SFMs is derived from tyrosine, alanine, glycine, and methionine (Figure 18) [115,116]. L-tyrosine undergoes *C*-methylation, oxidation, and *O*-methylation to afford the precursor 3-hydroxy-5-methyl-*O*-methyltyrosine (**152**) (Figure 19A) [117,118]. In 2010, Oikawa and co-workers reconstituted the formation of the core structure in vitro, revealing that two Pictet–Spengler (PS) reactions were involved in this process. Specifically, **152** was firstly activated and uploaded onto PCP of NRPS SfmC, assembling with intermediate **153** produced by NRPSs SfmA and SfmB via the first PS reaction. Following this, **154** formed by the subsequent reduction underwent a second PS reaction and reduction to generate **155** (Figure 19B) [119,120]. The following reduction and intramolecular cyclization of **155** yielded **156** which was then oxidized and methylated to generate **157**. Subsequently, **157** was transported outside the cell in company with its fatty acid chain and was removed by the membrane-anchored protein SfmE to produce **158**. Finally, SfmCy2 catalyzed the extracellular deamination of **158** to form **142,** indicating a prodrug maturation process (Figure 20) [121].

## 5. Others

### 5.1. Leinamycin

Antitumor agent leinamycin (LNM, **159**) is a hybrid peptide-polyketide NP and contains a unique 1,3-dioxo-1,2-dithiolane moiety, which is essential for its anticancer activity (Figure 21). The alkylation of DNA involves a rearrangement reaction in which LNM is initially activated via the attack of thiol to form the sulfenic acid intermediate **165**, which also enabled, triggering oxidative DNA cleavage via generating unstable hydrodisulfide intermediate (RSSH) (Figure 21) [122,123,124,125]. The oxathiolane intermediate **165** could be produced by breaking the S-S bond of LNM via thiol attack or hydrolysis. The C6-C7 alkene of **165** then attacks the electrophilic sulfur of the oxathiolanone group to generate an episulfonium ion intermediate, **166**, following an intermolecular nucleophilic attack with the N7 of guanine residues in duplex DNA to yield the DNA–drug adduct **168**. Unlike the reduction-mediated alkylation of DNA by LNM, LNM E1(**160**) could be transformed to its episulfonium ion intermediate **167** through an oxidative reaction catalyzed by reactive oxygen species (Figure 21) [126].

The skeleton of LNM was biosynthesized by hybrid NRPS-PKS assembly lines [127]. The LnmQ, LnmP, and NRPS module of LnmI are responsible for the unique thiazole-containing starter unit. The polyketide chain of LNM is elongated by two AT-less type I PKSs, LnmI and LnmJ, as well as a *trans*-AT enzyme LnmG [128,129,130,131,132]. A *β*-branched C3 unit derived from methylmalonyl-CoA was then installed by a set of proteins including a free-standing ACP (LnmL), a bifunctional AT-decarboxylase (LnmK), an HMGS homolog (LnmM), and an ECH homolog (LnmF) (Figure 22) [133,134]. Module 8 in LnmJ, containing a domain of unknown function domain (DUF), added an L-cysteine into C-3 of intermediate **170** to produce intermediate **171**. The PLP-dependent cysteine lyase domain (SH) can catalyze the cleavage of the C-S bond to yield **172,** which is cyclized to release the chain to generate LNM E1 (Figure 22) [135]. Recently, Meng et al. demonstrated that LnmJ-SH domain directly installed a -SSH group into the LNM polyketide scaffold via cleavage of the C-S bond linking the thiocysteine to form LNM E (**173**), which might be further transferred to LNM by a set of uncharacterized enzymes [136,137].

### 5.2. Gilvocarcins

Antitumor antibiotic gilvocarcins are a subfamily of *C*-glycoside aromatic polyketides and are derived from the typical angucycline scaffold [138]. Part of gilvocarcin-type natural products including gilvocarcin V (**174**), chrysomycin A (**175**), and ravidomycin (**176**) (Figure 23A) possess the vinyl substituent at C-8 which mediates a photo-activated [2 + 2] photocycloaddition with the thymidyl residue on DNA (Figure 23B) [139,140].

During the biosynthesis of **174**, the type II PKS and cyclases afford the angucycline precursor **177** starting from the starter unit propionyl-CoA [141,142]. Subsequent dehydration of **177** generates **178,** whose C-ring is oxidatively rearranged to yield **182** through possible intermediates **180** and **181**. Following this, **182** is glycosylated to form pregilvocarcin V, which is finally oxidized to gilvocarcin V (Figure 24) [143,144,145,146].

## 6. Perspective

Since DNA-alkylating NPs exhibit potent antitumor activity, their biosynthesis has received extensive attention. Obviously, the highly active functional groups including epoxide, cyclopropane, aziridine, and imine in their chemical structures play an important role in alkylating DNA to form the DNA–drug adduct. Elucidation of their biosynthetic pathways not only facilitates the discovery of new NPs with these biological active groups by genome mining but also is valuable for engineering NPs and drug design [147]. The unprecedented enzymology involved in their biosynthetic pathways can exert a positive influence on the development of biocatalysts as well. As a result of their strong DNA-alkylating activities, their producers have to confer self-resistance strategies to avoid damaging themselves, mainly through the cleavage of the DNA–drug complex or modification of the functional groups. Therefore, resistant gene-guided genome mining also contributes to discovering new DNA-alkylating antibiotics [148]. In addition, other moieties which contribute to affinity and reactivity with DNA are also indispensable for their ability to alkylate DNA. The modification of these moieties may enhance their activity or sensitivity and facilitate linking the NPs to antibodies via chemical synthesis.

## Figures and Tables

**Figure 1 molecules-27-06387-f001:**
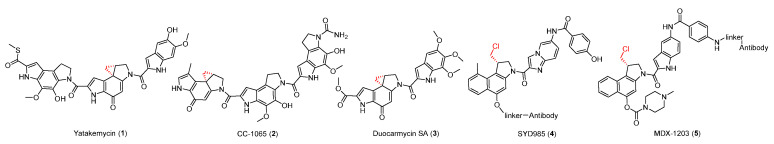
Chemical structures of spirocyclopropylcyclohexadienone family compounds.

**Figure 2 molecules-27-06387-f002:**
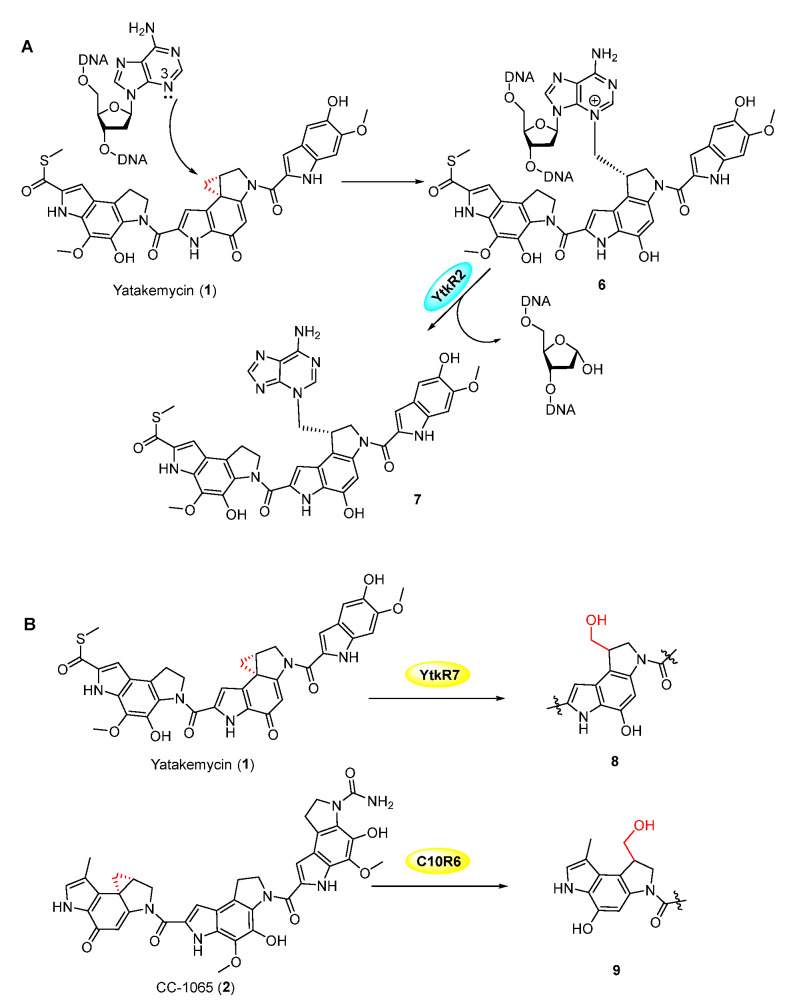
(**A**) DNA modification by YTM, and excision of DNA-drug complex by YtkR2. (**B**) Hydrolysis of the cyclopropyl moiety in YTM and CC-1065 by YtkR2 and C10R6, respectively.

**Figure 3 molecules-27-06387-f003:**
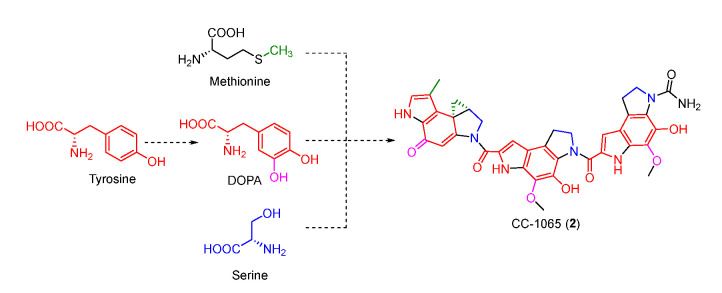
Isotopic labelling patterns with serine, methionine, and tyrosine.

**Figure 4 molecules-27-06387-f004:**
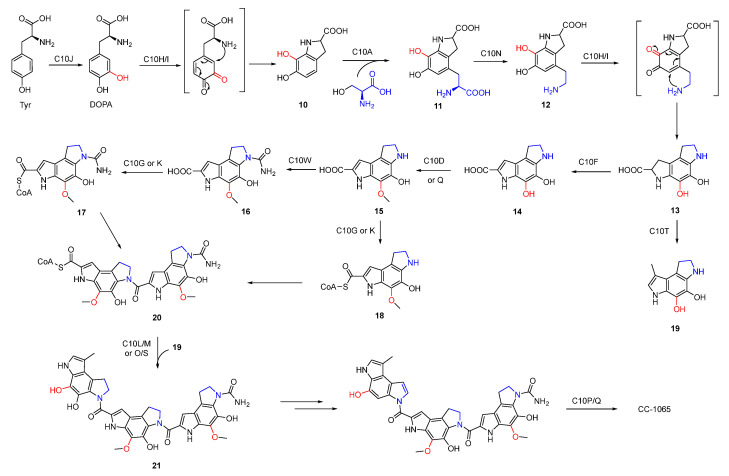
Proposed biosynthetic pathway of CC-1065.

**Figure 5 molecules-27-06387-f005:**
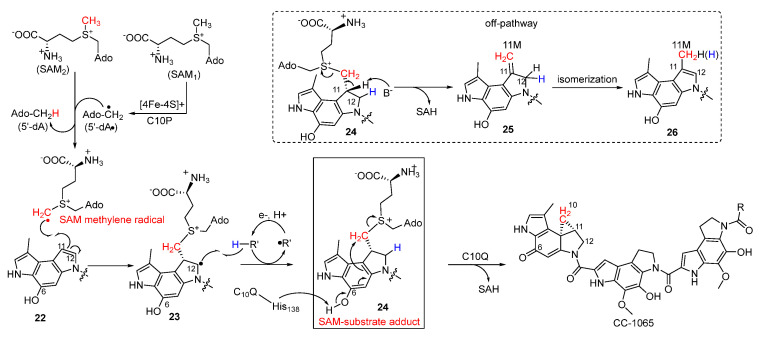
Proposed enzymatic mechanism of the cyclopropane moiety formation catalyzed by C10P and C10Q.

**Figure 6 molecules-27-06387-f006:**
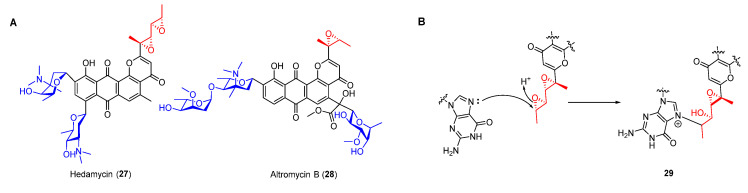
(**A**) Chemical structures of pluramycins. (**B**) Alkylation of DNA by hedamycin.

**Figure 7 molecules-27-06387-f007:**
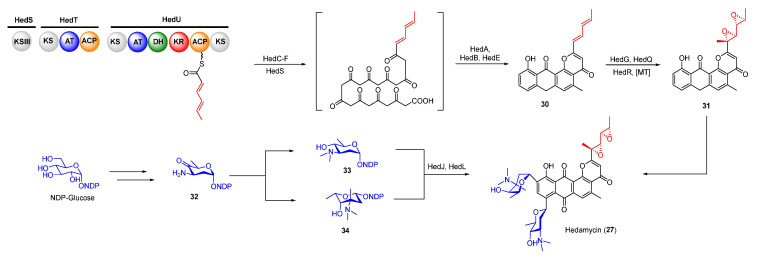
Proposed biosynthetic pathway of hedamycin. KS, ketosynthase; KR, ketoreductase; ACP, acyl carrier protein; DH, dehydratase; AT, acyl transferase.

**Figure 8 molecules-27-06387-f008:**
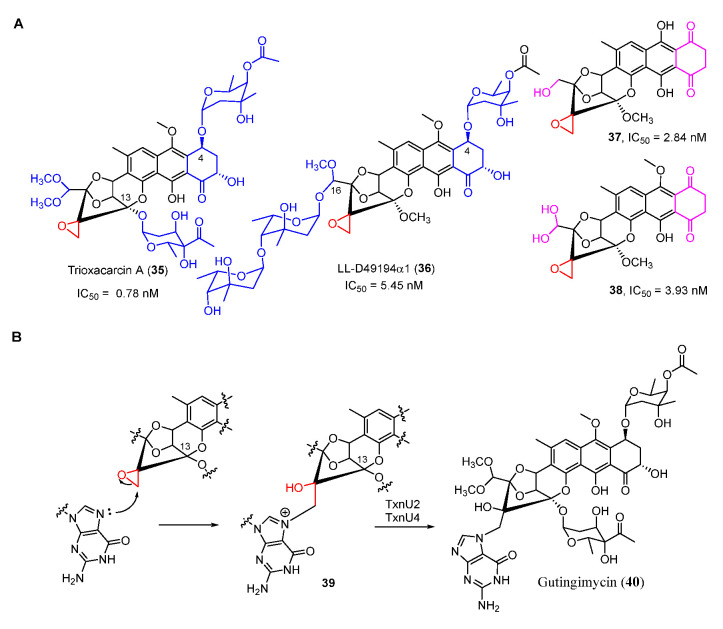
(**A**) Chemical structures of trioxacarcin A, LL-D49194α1 and their derivates. (**B**) Alkylation of DNA by trioxacarcins.

**Figure 9 molecules-27-06387-f009:**
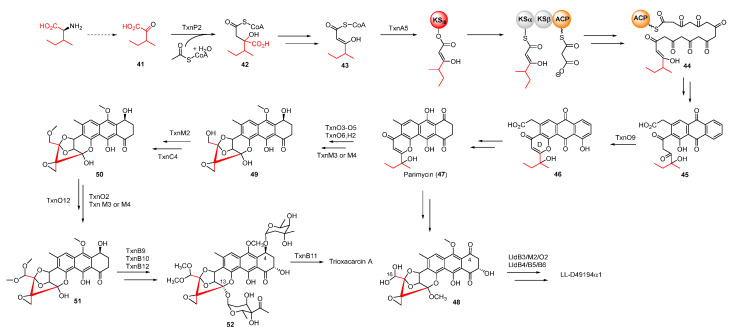
Proposed biosynthetic pathway of trioxacarcin A and LL-D49194α1. CLF, chain length factor.

**Figure 10 molecules-27-06387-f010:**
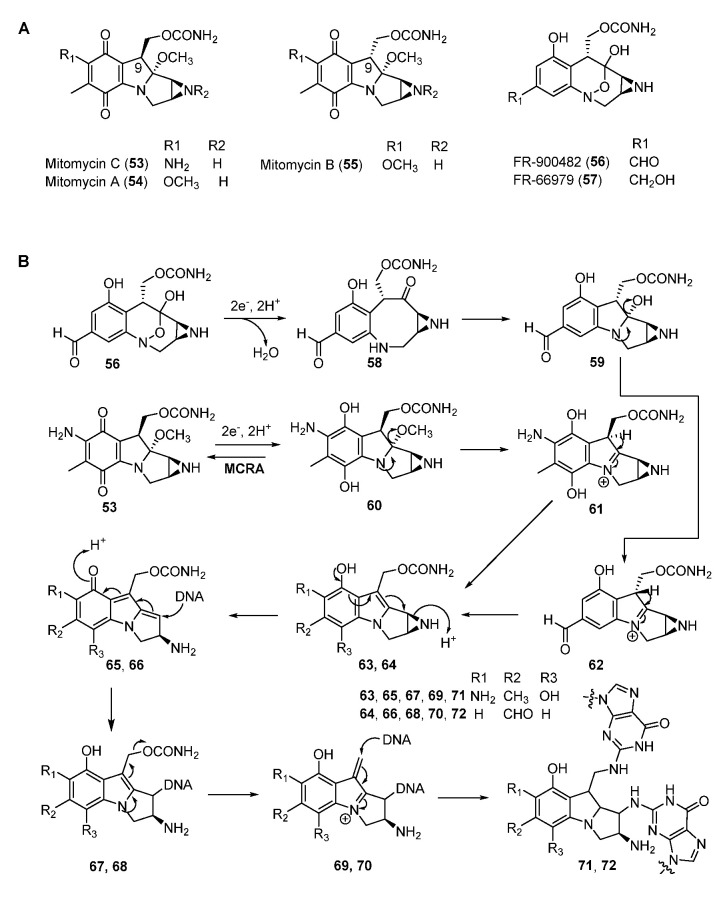
(**A**) Chemical structures of mitomycins. (**B**) Proposed mechanism of DNA cross-linking by mitomycin C and FR-900482.

**Figure 11 molecules-27-06387-f011:**
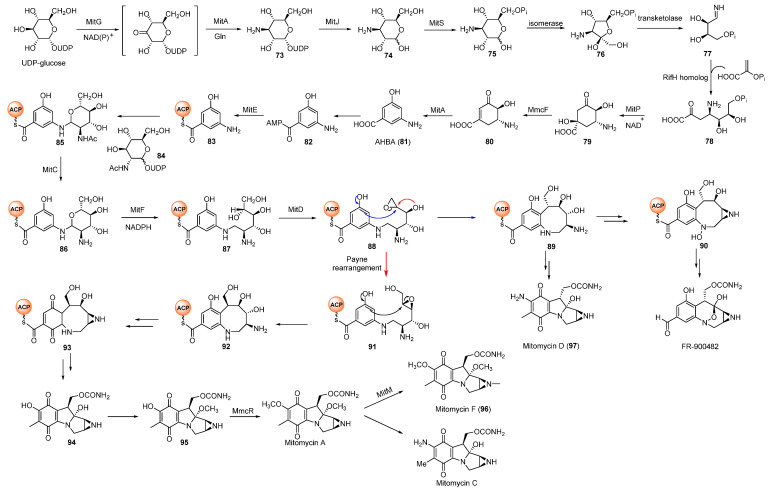
Proposed biosynthetic pathway of mitomycins.

**Figure 12 molecules-27-06387-f012:**
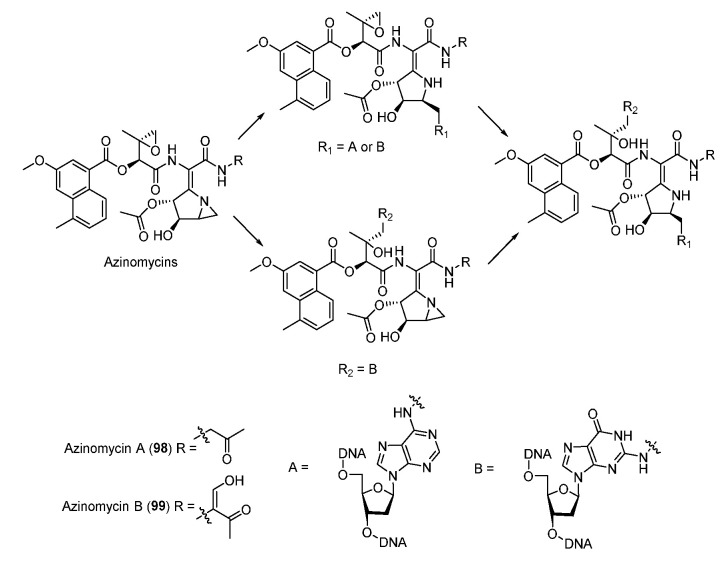
Proposed mechanism of DNA cross-linking by azinomycins.

**Figure 13 molecules-27-06387-f013:**
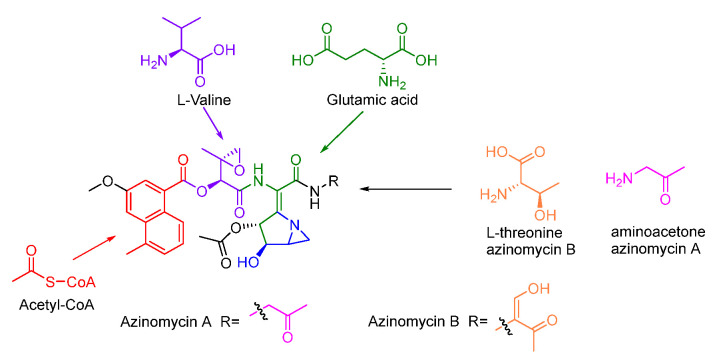
Origins of azinomycins revealed by isotopic labelling experiments.

**Figure 14 molecules-27-06387-f014:**
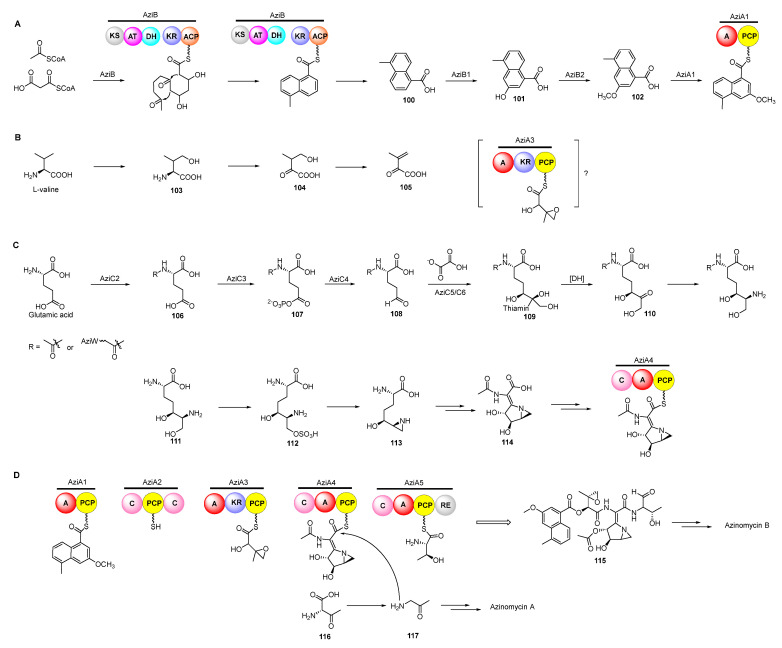
Proposed biosynthetic pathway of azinomycins. PCP, peptidyl carrier protein; A, adenylation; C, condensation; RE, reduction. Proposed pathway of constructing 3-methoxy-5-methyl-NPA moiety (**A**), epoxy intermediate (**B**), azabicyclic fragment (**C**), and incorporating building blocks (**D**).

**Figure 15 molecules-27-06387-f015:**
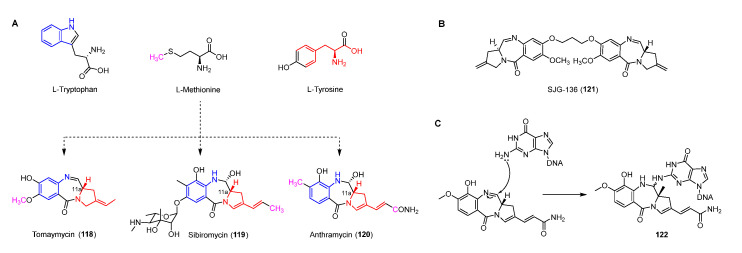
(**A**) Origins of pyrrolobenzodiazepines revealed by isotopic labeling experiments. (**B**) Chemical structure of SJG-136. (**C**) Proposed mechanism of DNA alkylating by pyrrolobenzodiazapines.

**Figure 16 molecules-27-06387-f016:**
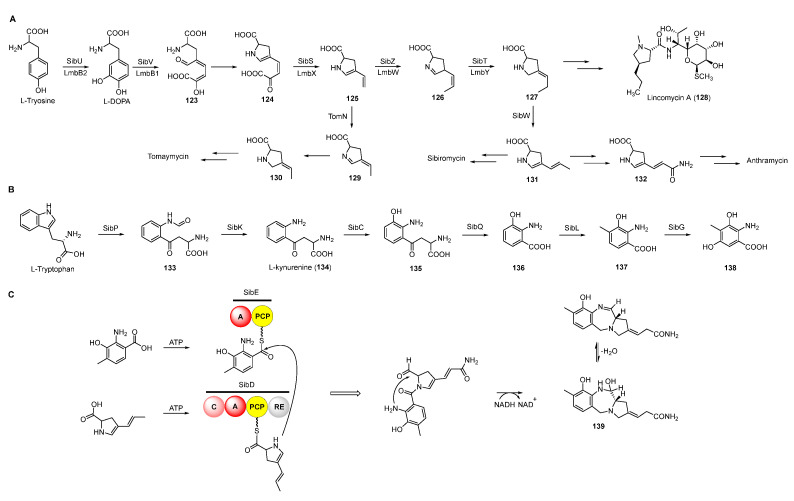
Proposed biosynthetic pathway of dihydropyrrole moieties (**A**), anthranilic acid moieties (**B**), and assembling two building blocks (**C**).

**Figure 17 molecules-27-06387-f017:**
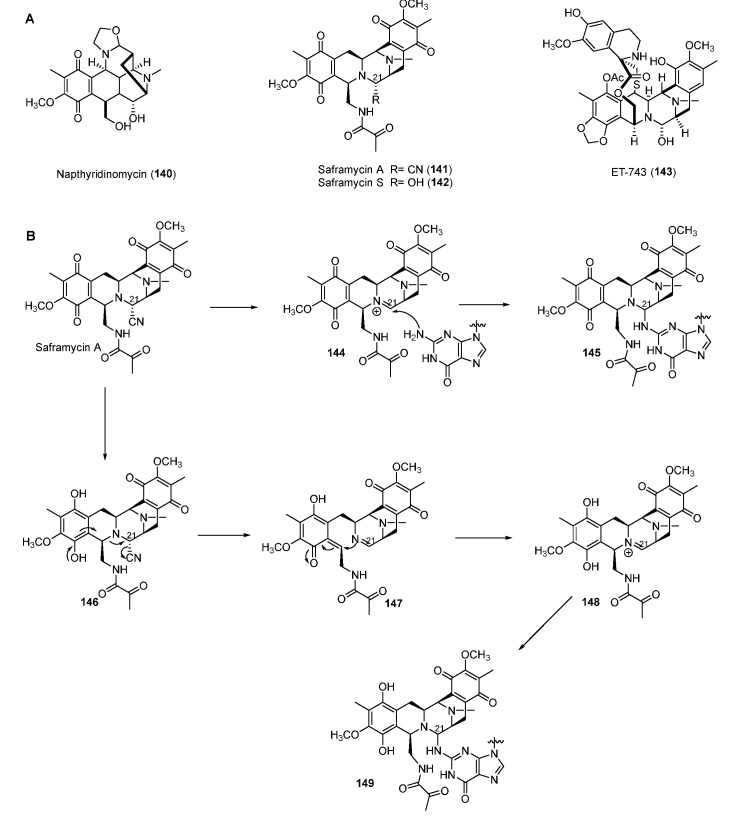
(**A**) Chemical structures of tetrahydroisoquinoline family. (**B**) Proposed mechanism of DNA alkylating by saframycin A.

**Figure 18 molecules-27-06387-f018:**
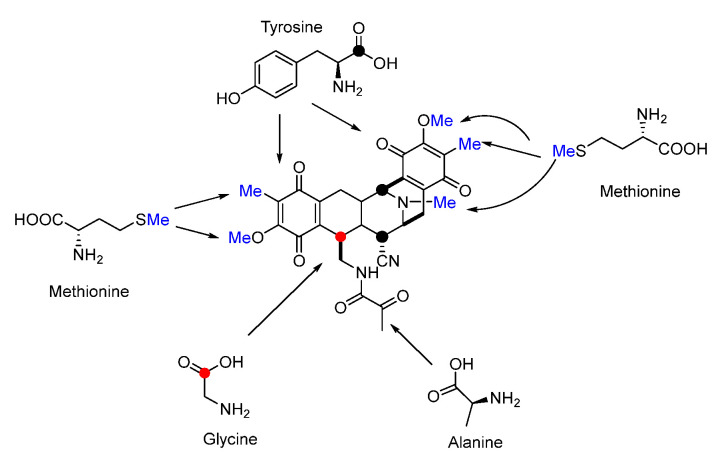
Origins of saframycins revealed by isotopic precursors labeling experiments.

**Figure 19 molecules-27-06387-f019:**
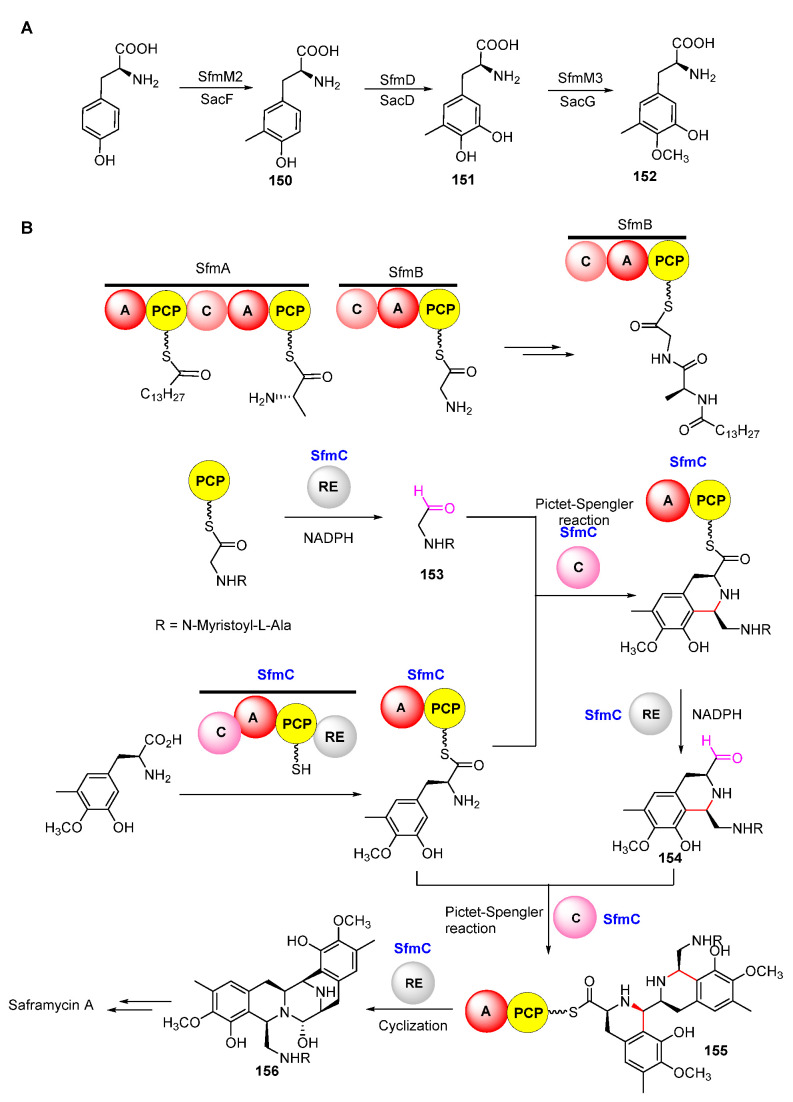
Proposed biosynthetic pathway of quinone moiety (**A**) and forming the skeleton of saframycins (**B**).

**Figure 20 molecules-27-06387-f020:**
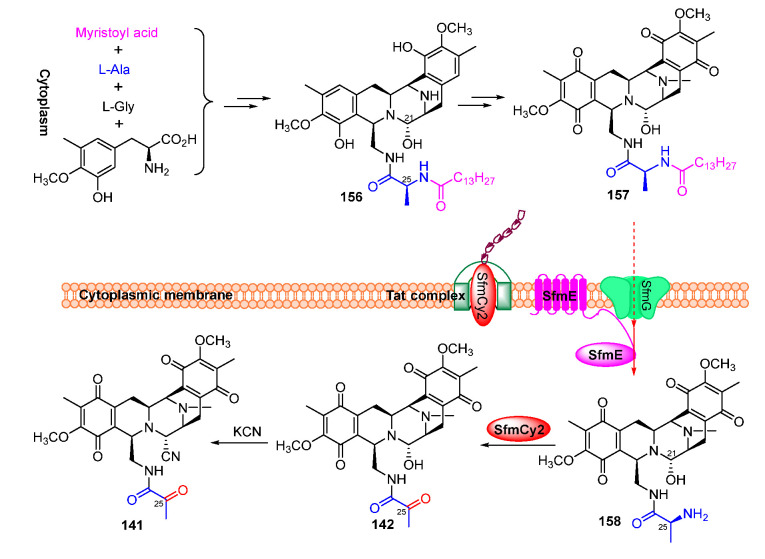
Post-modification of saframycins.

**Figure 21 molecules-27-06387-f021:**
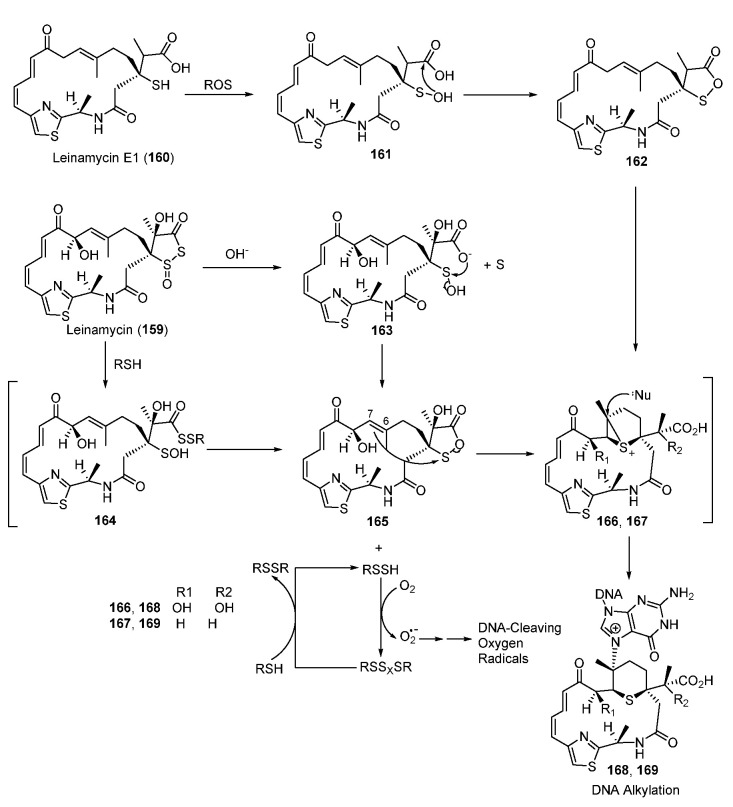
Proposed mechanisms of DNA alkylating by leinamycin and leinamycin E1.

**Figure 22 molecules-27-06387-f022:**
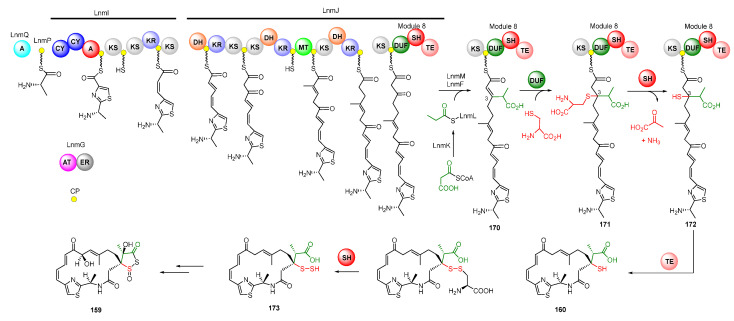
Proposed biosynthetic pathway of leinamycin. CP, carrier protein; CY, condensation/cyclization; MT, transferase; ER, enoyl reductase; TE, thioesterase; DUF, the domain of unknown function; SH, PLP-dependent cysteine lyase domain.

**Figure 23 molecules-27-06387-f023:**
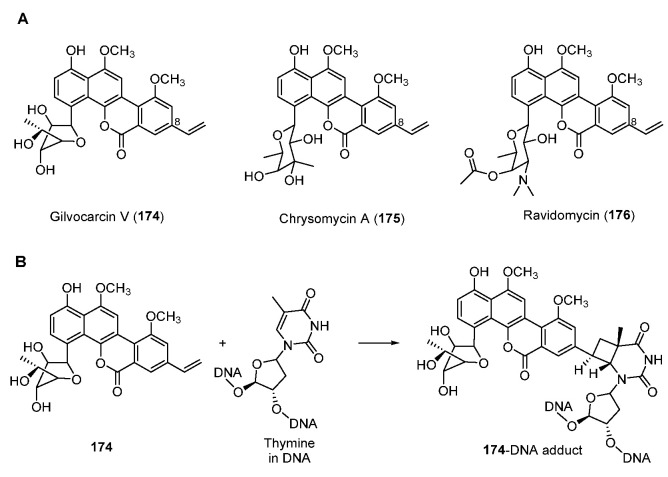
(**A**) Chemical structure of gilvocarcin V, chrysomycin A, and ravidomycin. (**B**) Mode of action of vinyl-containing gilvocarcin-type natural products.

**Figure 24 molecules-27-06387-f024:**
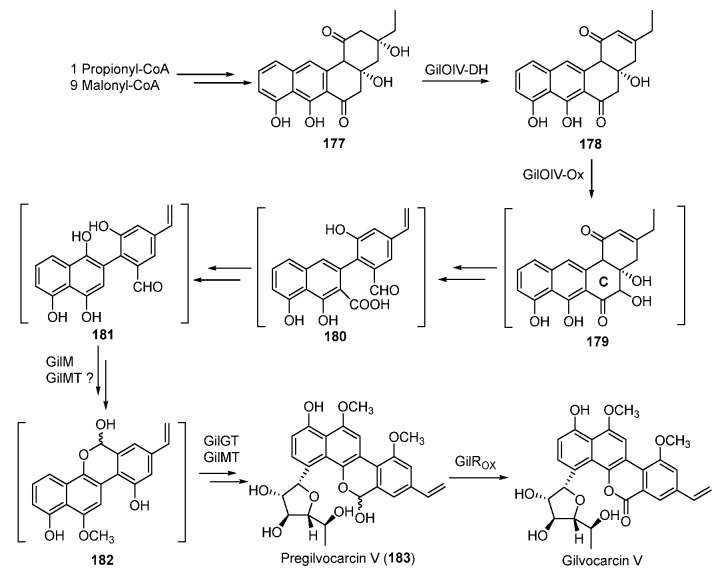
Proposed biosynthetic pathway of gilvocarcin V.

## Data Availability

Not applicable.

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
