# Peer review of "Biosynthesis of DNA-Alkylating Antitumor Natural Products"

_molecules, 2022, doi:10.3390/molecules27196387_

Round 1

Reviewer 1 Report

This manuscript summarized the research progress in DNA-alkylating antitumor natural products, focusing on the biosynthesis, modes of action, and auto-resistance mechanisms, providing readers with a more comprehensive understanding of these natural products. This manuscript is well written and organized.

Some minor concerns include:

(1)   L45, delete products;

(2)   L73, mechanism of the cyclopropyl moiety formation catalyzed by C10P and C10Q;

(3)   L97, change essential to important;

(4)   L109, delete starter, and type I PKS;

(5)   L119, change in to against;

(6)   Figure 9, the compound 42 should be attached to CoA;

(7)   L158, delete contain;

(8)   L164, delete Among;

(9)   L169, change went through further to underwent;

(10) Figure 14, some new progresses should be updated.

(11) L264, SJC136 is not consistent with Figure 15B;

(12) L276, some references are needed for including quinolobactin, daptomycin, and actinomycin; and change goes through to undergoes;

(13) L282, change afforded to “to afford 139”;

(14) Figure 16, For the last step, the thioester should be first reduced to the aldehyde group by the RE domain, and then condensed with the amino group to form the imine.

(15) L294 ET-743 and L297 Et-743 not consistent;

(16) L345, change yielded to “to yield the DNA-drug adduct 168”;

(17) For Gilvocarcins part, the all compound numbers are not bold;

(18) L376, delete unique because propionate as starter unit is common;

(19) Figure 24, change propionate and malonate to propionyl-CoA and malonyl-CoA, because malonate is present as malonyl-CoA in vivo.

Author Response

Following the suggestions, we changed all the nineteen points.

Reviewer 2 Report

Tang and co-workers nicely covered the review on Biosynthesis of DNA-alkylating antitumor natural products. In which they cover the DNA-alkylating antitumor natural products, including the biosynthesis, modes of action and auto-resistance mechanisms which is very interesting to reader and it's give good idea to understand the how the chemical structure especially the functional group play the vital role in the formation DNA drug molecules. I strongly recommend the manuscript to be published in the given form, however, I would suggest that the author should properly organize the space for chem draw and in some cases redraw the structure in chem draw and it should be aligned in the paper with words in the paragraph. 

Author Response

We optimized some structures.

Reviewer 3 Report

The manuscript submitted by Nie etal  reports the comprehensive review of the DNA-alkylating antitumour natural products, followed by their biosynthetic pathways and mode of action. The literature review was thorough and well-written. The authors make a systematic contribution to the research literature in this area of investigation.

Below are my minor comments,

1. Figure 12, please correct the compound name Azinimycins to Azinomycin in chemdraw.

2. Add compound numbers to Figure 23 B.

3. Figure 12: check the structure, this is not DNA-cross-linking (I don't see any DNA reaction).

4.  Figure 17B, please show the DNA attack on the reaction arrow.

5. Please ensure that references 3, 56 and 57 are formatted.

Author Response

We changed following the suggestions.